# Simultaneous interpreting with auto-subtitling: Investigating viewer cognitive effort, stress, and comprehension

Yanlin Li[1], Jiawen Diao[2], Andrew K. F. Cheung[2*]

1 Department of Industrial and Systems Engineering, Hong Kong Polytechnic University, Kowloon, Hong Kong, 2 Department of Chinese and Bilingual Studies, Hong Kong Polytechnic University, Kowloon, Hong Kong

* andrew.cheung@polyu.edu.hk

## Abstract

Simultaneous interpreting (SI) enables real-time cross-language communication without significant delays and is vital for fast-paced environments such as multilingual conferences. Automatic subtitles, powered by artificial intelligence (AI), is an important mode of audiovisual translation and has been widely deployed by virtual conferencing platforms to help users overcome language barriers. While the cognitive and emotional impacts of SI have been explored in prior studies, research directly comparing SI, auto-subtitling, and their combined use remains limited. This study investigates and compares the effectiveness of three interlingual translation modes, auto-subtitling, SI, and a combined dual-modality approach, on comprehension, cognitive load, and stress levels. Mandarin Chinese-speaking participants viewed a video presentation delivered in Arabic, a language they did not understand. Participants were divided into three groups: Group A relied on automatic subtitles in Simplified Chinese characters, Group B relied on SI in Mandarin, and Group C used a combination of both methods. Analysis of electroencephalographic data and comprehension test results revealed no statistically significant differences in content comprehension across the groups. However, Group A reported the poorest viewing experience, with the highest stress levels, while Group B expended the greatest cognitive effort. Group C exhibited the lowest levels of cognitive effort and stress, underscoring the advantages of dual-modality systems. These findings suggest that combining accurate automatic subtitles with professional interpreting may enhance accessibility, reduce cognitive demands, and improve the viewing experience, offering valuable insights into the integration of AI-driven technologies in SI.

## Introduction

The purpose of this research is to experimentally examine the effects of three interlingual modes of translation, i.e., auto-subtitling, simultaneous interpreting (SI), and

**Data availability statement:** All relevant data are within the manuscript and its Supporting information files.

**Funding:** We would like to thank the Research and Innovation Office of the Hong Kong Polytechnic University for supporting the project (Project Code: RKQY). This manuscript was also partly supported by funding from Project number P0046386 of the Department of Chinese and Bilingual Studies of the Hong Kong Polytechnic University. The funders had no role in study design, data collection and analysis, decision to publish, or preparation of the manuscript.

**Competing interests:** The authors have declared that no competing interests exist.

a combination of both (dual-modality), on viewer comprehension, cognitive load, and viewing experience during multilingual presentations. Specifically, the study evaluates whether automatic subtitles, generated by artificial intelligence (AI), can facilitate content comprehension comparably to SI, analyzes the cognitive demands experienced by viewers across these three conditions, and identifies the most effective mode for cross-linguistic communication. By systematically addressing these aspects, this research seeks to enhance audience understanding and engagement in multilingual settings.

The rise of multimedia communication has driven the development of audiovisual translation (AVT), encompassing subtitling, dubbing, and voiceover. Among these, subtitling is one of the most widely adopted due to its cost-effectiveness and efficiency [1,2]. Originally developed to facilitate access for deaf/hard-of-hearing audiences and those with limited proficiency in the source language, subtitling now serves critical functions in both educational and entertainment contexts [3–5]. In language education, subtitles have been shown to support vocabulary acquisition, listening comprehension, and learner engagement by providing concurrent visual and auditory input [6]. Research further highlights their role in long-term language development, particularly through autonomous exposure to captioned media (e.g., films, TV programmes) in target languages [7]. In the entertainment industry, subtitling enables cross-cultural access to audiovisual narratives, allowing global audiences to engage with original-language performances, a function amplified by streaming platforms like Netflix, YouTube, and TikTok [8].

Traditionally, real-time subtitling relied on stenotyping or skilled subtitlers using dual keyboards [1]. These labor-intensive methods limited scalability until the advent of AI-driven solutions. Automatic speech recognition (ASR) has transformed subtitling practices. ASR, which converts spoken language into written text using algorithms, has enabled the adoption of real-time automatic subtitles, also known as live captioning [9]. These systems are embedded in platforms like Zoom and Microsoft Teams, providing near-instantaneous subtitles to enhance accessibility for global audiences [8]. While ASR technologies now enable real-time automatic subtitles in general contexts, challenges persist in specialized, high-stakes settings, such as multilingual conferences, where SI is traditionally employed.

SI is a cornerstone of multilingual communication, enabling real-time exchange across linguistic boundaries in international and institutional contexts [1,2]. This mode of interpretation relies on highly skilled human interpreters who deliver accurate and immediate renditions of the source language into the target language [9]. While effective, professional SI services are resource-intensive, requiring significant human and financial investment [10]. The pandemic-driven shift to virtual and hybrid conferencing has amplified these challenges, exposing the limitations of traditional SI services in terms of scalability and affordability [11,12]. Although remote SI has emerged as a solution, it remains constrained by interpreter availability and cost. Recent advancements in auto-subtitling offer a complementary alternative. These systems provide real-time textual support to enhance accessibility in formal, multilingual conference settings, such as academic symposia and international policy forums

[10]. AI-powered tools like Otter.ai and Google Meet's live captioning are increasingly utilized to support multilingual communication in these contexts.

Existing research has extensively examined the advantages and limitations of subtitles across various contexts. Automatic subtitles, created using ASR technology, have been shown to enhance comprehension by bridging auditory and textual input, improving immersion, and maintaining viewer focus [13–17]. However, such advantages are not universal. When subtitle speed is high, content is dense, or synchronization is poor, the sequential processing of visual (textual) and auditory information may actually increase cognitive effort, potentially undermining comprehension [18,19].

This study focuses specifically on open (burned-in) interlingual automatic subtitles, machine-generated translations embedded directly into video, used in real-time or pre-recorded multilingual communication. A proposed enhancement in such settings is the dual-modality approach, which combines SI with auto-subtitling to distribute information across both auditory and visual channels [16]. While this multimodal design holds promise for supporting diverse processing preferences and compensating for gaps in either channel, it may also impose greater cognitive demands, as viewers are required to shift attention between listening and reading, particularly in high-stakes or fast-paced scenarios [11,20]. While auto-subtitling systems have improved, errors in transcription or translation persist, particularly in noisy audio or linguistically complex content [20].

While studies have explored the integration of ASR tools in interpreting workflows, they primarily focus on the impact on interpreters' performance rather than the viewers' experience with dual-modality systems [9]. For instance, Defrancq and Fantinuoli [21] highlight the psychological benefits of ASR availability for interpreters, while Tammasrisawat and Rangponsumrit [22] demonstrate that ASR-computer-assisted interpreting (CAI) tools can reduce error rates and omissions, improving interpreters' terminology rendition quality. However, little is known about how the combination of automatic subtitles and SI affects viewers' comprehension, cognitive load, and overall experience. It remains unclear whether this approach enhances understanding or introduces new challenges due to the cognitive demands of processing dual modalities.

This gap in the literature underscores the need for a rigorous investigation into the combined effect of auto-subtitling and SI on viewers. Such research can clarify whether the dual-modality approach strikes the right balance between accessibility and cognitive demands, offering practical insights for improving multilingual communication. To address this gap, we propose the following research questions (RQs):

**RQ1:** How do automatic subtitles compare to SI in facilitating content comprehension for viewers?

**RQ2:** What differences in cognitive demands do automatic subtitles and SI impose on viewers?

**RQ3:** How do automatic subtitles and SI influence viewers' overall viewing experience?

The contributions of this study are twofold. First, it examines how audiences cognitively respond to dual modality input, which is the combination of SI and automatic subtitles, by evaluating comprehension, cognitive effort, and viewing experience in multilingual conference settings where real-time processing is essential. While prior research has focused on interpreters' performance or single-modality delivery, this study addresses a notable gap by providing empirical analysis of audience experience in dual-modality environments, offering insights in AVT. Second, the study enhances methodological rigor by employing electroencephalographic (EEG) techniques to objectively measure viewers' cognitive load. This approach moves beyond self-reported data to reveal how cognitive resources are allocated during dual-modality processing.

## Literature review

### Automatic subtitles and the viewing experience

Traditional subtitles serve as textual representations of spoken information in videos, fulfilling a variety of essential functions. Initially created to provide translations by human translators for movies and television audiences facing language barriers, subtitles have evolved into indispensable tools for language learning, cognitive load management, and improving viewer comprehension [18,23–25].

Automatic subtitles are generated by ASR technology that enables the transcription of spoken language into text by analyzing the shape of speech waves [26]. Unlike traditional subtitles, automatic subtitles are generated in real time, adapting to the speaker's pace and often employing innovative formats such as multi-line stacking to improve readability [27]. Auto-subtitling has gained traction for its efficiency and scalability, particularly in environments requiring immediate content delivery, such as live broadcasts, online learning platforms, and virtual meetings [21].

ASR has undergone significant advancements since its inception. In 1952, Davis et al. [28] pioneered early speech recognition systems capable of recognizing digits, achieving notable accuracy despite the technological limitations of the time. Modern ASR systems, powered by AI and neural network architectures such as the Transformer model, have since revolutionized speech-to-text technology, enabling real-time transcription of large-scale audio data with greater accuracy and reliability [29]. However, accuracy in ASR remains a multifaceted challenge, particularly in real-world environments. This includes challenges in recognizing speech accurately in the same language (monolingual transcription) and auto-translating speech into another language (interlingual transcription) [9,16,17,30]. Factors such as background noise, accents, dialects, and linguistic variations further complicate recognition tasks [29,30]. These limitations are particularly critical in high-stakes settings such as live conferences, where errors in recognition or translation could lead to miscommunication or misunderstandings.

Despite these challenges, automatic subtitles have become an essential tool for facilitating accessibility. They are especially valuable for ensuring accessibility for deaf or hard of hearing individuals, as well as for native [31] and non-native speakers [32], including those in noisy or quiet environments, multitaskers, and others with specific needs or preferences. Platforms such as Zoom and Microsoft Teams have integrated auto-subtitling features as part of their services to address accessibility needs. Although these features are typically included in paid licenses, they are perceived as lower-cost alternatives to human-provided services, such as SI or professional subtitling, due to the scalability and automation of ASR technology [20]. Auto-subtitling might not be as effective as having SI because it lacks the nuances of spoken language that help keep people interested and oriented [33]. Therefore, some may prefer overcoming language barriers by listening to SI because the need to expend excessive cognitive effort to follow visually a narrative may result in a negative viewing experience [34].

Research to date has focused on the impact of automatic subtitles on interpreters themselves [9,21,22]. These discussions have largely emphasized how auto-subtitling can support interpreters by improving their accuracy and reducing cognitive load [35]. However, listeners can also benefit from the integration of automatic subtitles. In virtual conferences, where there is a need to accommodate diverse linguistic requirements and ensure accessibility for a broader audience, automatic subtitles can play a crucial role in enhancing the delivery of multilingual content. This is particularly important in settings with participants from varied linguistic and cognitive backgrounds, where balancing visual and auditory inputs is essential for optimizing information accessibility [10].

## Measuring cognitive effort and viewing experience using EEG data

As auto-subtitling systems become increasingly prevalent across various contexts [3,21,22,36], understanding their impact on viewers' cognitive performance and comprehension is crucial for optimizing communication and reducing language barriers [19]. The concepts of cognitive effort and stress may appear similar but are in fact distinct. Cognitive effort is typically defined as the mental resources allocated for processing a task [36], whereas stress refers to a particular relationship between an individual and their environment in which the demands of the situation are perceived as taxing or exceeding the individual's resources, potentially affecting their focus and functional performance [37]. In the context of viewers' audiovisual content comprehension, cognitive effort relates to the mental resources required to process information when viewing automatic subtitles and simultaneously listening to spoken content [19]. Stress, on the other hand, arises when the cognitive demands of processing audiovisual information are perceived as overwhelming or exceeding an individual's comfort level [37]. This can affect viewers' ability to maintain focus and process content effectively, especially

in high-pressure environments such as live conferences. Research into viewers' cognitive performance in audiovisual content comprehension, particularly in the context of automatic subtitles, represents a vital area of study. It investigates the relationship between multimodal content processing and cognitive functions, providing valuable insights into how technologies like auto-subtitling can either support or obstruct comprehension, ultimately shaping the effectiveness of information delivery.

To effectively measure cognitive effort and stress, researchers commonly utilize both subjective and objective assessment methods. Subjective measures, such as self-report questionnaires, enable participants to describe their perceived levels of cognitive effort and stress during specific tasks [38,39]. For instance, self-reported data in [40] suggested that computer-assisted consecutive interpreting resulted in fewer pauses, reduced cognitive load, and improved overall interpreting quality, particularly in terms of accuracy. However, conflicting findings have emerged. The conflict matrix, which lacks "reliable empirical data to substantiate" its claims [41], suggests that while additional information might logically enhance interpretation accuracy, comparisons between cognitive resource allocation and the conflict matrix for SI with and without text predict a discrete increase in task interference, and consequently, cognitive load. These conflicting findings may, in part, be attributed to the limited accuracy and reliability of subjective measures [41].

Objective measures offer a detailed understanding of cognitive effort and stress by tracking real-time physiological responses linked to cognitive processes. These responses usually involve the central and peripheral nervous systems [42]. The most common way to accurately decode the rhythm of the nervous system is to use EEG data [43], which enables researchers to correlate specific brain responses with performance metrics. In recent years, advancements in human–computer interaction and the increased accessibility of low-cost consumer-grade EEG devices have led to widespread recognition within research communities that objective EEG data provide a robust and reliable means of measuring participants' cognitive performance and the user experience [9,44–47].

Analyzing brain wave patterns is essential for understanding cognitive effort and stress levels [45]. While the relationship between alpha wave fluctuations and cognitive workload remains somewhat unclear [48], neural oscillations associated with theta and beta waves are recognized as reliable markers of cognitive workload [49]. According to previous studies [42,50], theta waves (4–8 Hz) are associated with memory processing and cognitive load; increased theta activity may indicate significant cognitive effort and greater strain on working memory [50] and increased mental fatigue [51–53]. In such a setting, theta power can indicate mental resource allocation during multimodal (audiovisual) content processing. Higher theta activity in the frontal region corresponds to executive functions, including decision-making, working memory, and semantic processing, while occipital region activity reflects visual processing and reading comprehension [54]. Regarding beta waves, Lazarus and Folkman [55] identified elevated beta wave power as an indicator of increased stress levels, as beta waves are associated with heightened cognitive activity and emotional strain during task performance. In particular, high beta wave activity often reflects mental states characterized by tension, anxiety, or demands associated with sustaining attention under pressure. These findings are further supported by research in interpreting and language translation contexts, which has used beta wave activity to measure cognitive and emotional strain [51,52].

The adoption of EEG technology has been instrumental in observing the cognitive mechanisms of simultaneous interpreters. Early studies by Kurz [56,57] explored EEG changes during SI, providing valuable insights into the mental processes underlying complex verbal tasks. These studies also confirmed the utility of computer-assisted neurophysiological measures for investigating cortical processes during SI. Building on this foundation, Koshkin et al. [58] employed EEG techniques to examine interpreters' neural activity under varying levels of working memory load during the SI of continuous prose. Similarly, Moser-Mercer [59] underscored the importance of studying cognitive functioning in simultaneous interpreters, highlighting the "plasticity" of the interpreter's brain and how experienced professionals can overcome common cognitive constraints through expertise. Yagura et al. [60] further advanced this area by quantifying differences in brain activity patterns between experienced and novice interpreters using EEGs in realistic SI environments. Their findings revealed that years of SI experience significantly influence selective attention during interpretation.

While existing studies primarily focus on the use of EEG techniques to investigate simultaneous interpreters' cognitive processes, research on viewers' cognitive performance under different interpreting modes remains sparse. Specifically, the combined effects of auto-subtitling systems and SI on viewers' cognitive performance have yet to be thoroughly explored. Given the growing emphasis on leveraging advanced technologies to enhance information accessibility, it is crucial to examine how these modes interact. This knowledge gap is particularly significant in light of the increasing integration of technology into interpreting contexts. Addressing this gap is essential not only for advancing theoretical understanding of interpreting practices but also for developing evidence-based approaches to improve interpreting performance and enhance user satisfaction in multilingual communication settings.

## Method

This study adopted a mixed-methods approach to examine participants' cognitive performance and comprehension. Participants' comprehension was assessed using an online questionnaire, while an EEG device measured their cognitive performance.

Ethics approval was obtained from The Hong Kong Polytechnic University Institutional Review Board (PolyU IRB, reference number HSEARS20241022005, 28 October 2024). Participants, recruited via offline classes, provided oral consent after reviewing a clear, non-technical information sheet outlining the study's aims and procedures. They were assured of their right to withdraw at any time without penalty.

## Participants

A total of 45 participants (28 female, 17 male) took part in the study, with 15 individuals in each of the three experimental groups (see Table 1 for group-specific demographics). All participants were native speakers of Mandarin Chinese and reported using Mandarin as their primary language of communication. Self-assessed proficiency aligned with the C2 level of the Common European Framework of Reference for Languages (CEFR). Screening questions confirmed no history of visual, auditory, or language-related impairments.

Participants were born and raised in mainland China, with representation from multiple provinces and municipalities, including Beijing, Shanghai, Guangdong, and Sichuan. They were recruited from a range of academic disciplines to ensure variation in educational background and to minimize domain-specific bias. None of the participants reported any academic or professional background in fields such as international relations, diplomacy, or United Nations (UN) affairs, based on the information gathered through background questionnaires. Although participants were not directly asked about their prior familiarity with the topic of the stimulus video, their disciplinary profiles and lack of relevant training suggest limited exposure to the domain. This helps to mitigate the influence of domain expertise on task performance and ensures that the study primarily captures general audience comprehension.

To examine the effects of different content delivery methods on cognitive effort, viewing experience, and comprehension, the 45 participants were randomly assigned to three experimental groups of equal size (n = 15): Group A (automatic subtitles), Group B (SI), and Group C (a combination of automatic subtitles and SI).

**Table 1. Demographic characteristics of participants (N = 45).**

| Demographic Characteristics | Total (N = 45) | Group A | Group B | Group C |
|---|---|---|---|---|
| Female | 28 | 10 | 9 | 9 |
| Male | 17 | 5 | 6 | 6 |
| Average Age | 23.7 | 23.5 | 23.6 | 23.9 |
| Age Range | 21-26 | 21-25 | 21-26 | 21-26 |

While stratified randomization was not employed at the assignment stage, this decision was informed by the modest sample size and logistical considerations. Given that demographic variables were not expected to exert strong effects on the dependent measures, simple randomization was deemed sufficient. Post-allocation checks confirmed satisfactory balance across key participant characteristics. Gender distribution was comparable across groups (Group A: 10F/5M, Group B: 9F/6M, Group C: 9F/6M), and participant ages ranged from 21 to 26 years, with group averages between 23.5 and 23.9 (see Table 1). Participants were also diverse in their academic backgrounds, with no concentration of disciplinary expertise in any one group. These checks support the internal validity of the group comparisons and help rule out confounding effects from participant-level variables.

The formal experiment commenced on November 1 and concluded on November 18, 2024. All participants wore EEG devices while watching the conference video in individual rooms without any intervention. We monitored their EEG signals from a separate independent room. To control for potential stress-inducing factors, the experimental environment was designed to be quiet and low-pressure. Tasks were untimed, and no performance-based feedback was provided. Participants were encouraged to complete the questionnaire at their own pace. Baseline measures of cognitive state were also collected to account for individual differences in initial stress levels. Informal debriefings indicated that participants generally did not perceive the task as stressful. These measures help ensure that any observed effects are attributable to the experimental manipulation.

### EEG headset and key brain waves

The Emotiv EPOC X 14-channel wireless EEG headset was used to record brain activity, with raw EEG data accessed through the Emotiv Pro Suite installed on an HP Pavilion Plus laptop. The headset was paired with Emotiv Pro via Bluetooth, enabling effective real-time monitoring of EEG data. The Emotiv headset provides the power of five frequency bands, namely theta (4–8 Hz), alpha (8–12 Hz), low beta L (12–16 Hz), high beta H (16–25 Hz), and gamma (25–45 Hz), for each sensor, as shown in Fig 1.

### Materials

A single speech in Arabic delivered at a UN conference was selected as the experimental material. This speech was chosen because it reflects the UN's standard practice of providing SI services into its six official languages, aligning with our focus on evaluating comprehension of interpreted content in professional settings. The original speech was interpreted into Chinese by professional UN interpreters, and the official video and audio recordings were publicly available on the UN website. To ensure experimental control, all participants across the three groups were exposed to identical speech content, which was then processed into three distinct delivery modes:

Version A: Chinese auto-subtitling generated by an ASR system based on the Chinese SI audio.

Version B: Chinese SI audio only, without subtitles.

Version C: A combination of Chinese SI audio and Chinese automatic subtitles.

The speech content, pacing, and audiovisual quality were held constant across all three versions. Each version lasted exactly 3 minutes and 40 seconds. The use of a single speech across all experimental conditions was an intentional design choice aimed at maximizing internal validity. This design ensured that the only manipulated variable was the mode of delivery, while the linguistic content, discourse topic, and communicative context remained constant. Using a single speech allowed us to isolate the effects of delivery mode on cognitive effort and comprehension outcomes without introducing variability from message content or situational features.

To assess the participants' comprehension of the speech in the target language, a multiple-choice test was developed in Simplified Chinese characters. The test consisted of 10 single-choice questions focused on factual recall, such as the

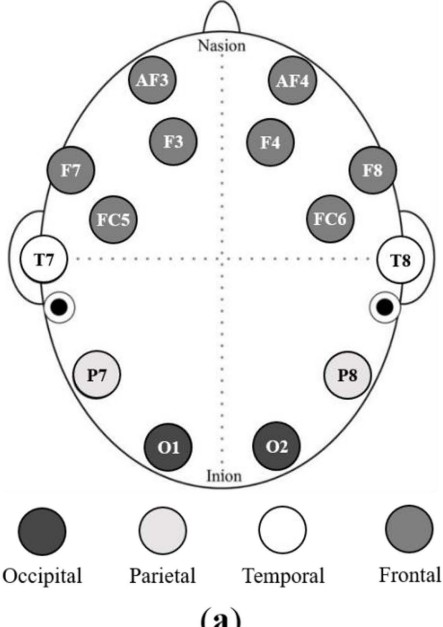

| Brain Wave | Frequency | Location |
|------------|-----------|----------|
| Delta | 0.5-3 Hz | Frontal region |
| Theta | 4-7 Hz | Parietal and temporal regions |
| Alpha | 8-13 Hz | Occipital region |
| Beta | 14-30 Hz | Parietal and frontal regions |

(a)                                                                 (b)

**Fig 1. Electrode placement of the Emotiv Epoc X EEG headset.** AF: anterior–frontal, F: frontal, FC: fronto-central, T: temporal, P: parietal, O: occipital. The odd numbers denote the left hemisphere and even numbers represent the right hemisphere. Black dots indicate locations of reference electrodes.

speech's main topics, numerical data, and named entities, to evaluate the participants' immediate understanding of the content. An example of a question translated in English is as follows:

"*What is the purpose of holding this summit?*

A) To promote global economic recovery

B) To emphasize national sovereignty and integrity

C) To enhance economic cooperation among countries

D) To reshape the multilateral system"

Correct answer:

D) This option directly reflects the central goal as stated in the speech: "reshape the multilateral system".

Distractor rationale:

A) Reflects general UN themes related to development and recovery but does not align with the summit's explicitly stated focus.

B) Draws on the principle of respecting sovereignty mentioned in the speech but misrepresents it as the main purpose.

C) Suggests international cooperation, a peripheral theme, but inaccurately elevates it to the summit's core objective.

Each question was worth 10 points, with no partial credit awarded for incorrect or incomplete answers. As a result, the total comprehension score ranged from 0 to 100, with higher scores indicating better comprehension. To ensure the test's

validity and reliability, all questions and answer options were reviewed by a linguistics expert who specialized in translation and interpreting. This review ensured that the questions aligned with the speech content and that the wording of the answer options was clear, precise, and appropriate for assessing comprehension of the target language.

## Experimental procedure

A pilot test was conducted prior to the main study to ensure the feasibility of the experimental design and procedures. This included evaluating the functionality of video playback, the clarity of the instructions, and the timing of the comprehension test. Feedback from the pilot phase informed minor adjustments, including the rewording of comprehension questions to improve clarity and technical refinements to ensure the smooth functioning of all experimental components. One such revision involved replacing the original item *"Why was the summit important?"*, which invited subjective interpretation, with *"What is the purpose of holding this summit?"*, a fact-based question more directly anchored in the explicit linguistic content of the source text. The pilot confirmed the feasibility of the materials and procedures for the main experiment. Accordingly, the formal study adopted the validated setup—including standardized video playback procedures—and was conducted in a controlled lab environment to ensure consistency and data reliability.

Participants were randomly assigned to one of three experimental conditions—SI, automatic subtitles, or dual-modality (SI + automatic subtitles)—using a computer-generated randomization sequence. Each participant experienced only one modality condition and viewed the same stimulus video in a fixed order, followed by comprehension questions. Random allocation was employed to control for potential between-group variation unrelated to the experimental treatment.

Each participant selected a session time based on their availability and was informed in advance that they would receive a HK$50 gift card as compensation. Based on each participant's selected schedule, the research team reserved the lab and ensured the appropriate condition materials were prepared. To minimize bias, participants were explicitly instructed not to discuss their experiences or the test content with one another until all sessions were completed.

Upon arrival, the participant was provided with a consent form to read and sign, ensuring ethical compliance with the study protocol. Following this, the participant was briefed on the experimental procedure and introduced to the wireless EEG device, which was used to monitor their cognitive activity. Each experimental session was conducted individually. A trained research assistant assisted the participant in correctly positioning the wireless EEG headset and ensured that 100% contact quality was achieved before proceeding with the task.

Once the EEG setup and baseline signal confirmation were complete, the participant was seated at an individual station and the video presentation commenced. During the video viewing, researchers monitored the participant from a separate room to ensure uninterrupted EEG data collection and to quickly address any technical issues. The environment was carefully controlled to eliminate distractions, such as noise or external interference, ensuring consistent testing conditions for each participant.

To maintain consistency, the video could not be paused or replayed during the experiment. The comprehension questions were only accessible after the video had ended, to ensure that the participant relied solely on the presented material. Fig 2 presents the participant wearing the EEG device while watching the video material.

# Results

## Content comprehension

After all participants completed the experiments, their comprehension scores were collected and processed for statistical analysis. To compare comprehension scores across the three groups, the Kruskal-Wallis test was employed. The analysis yielded an *H* statistic of 0.407 ($df = 2$, $p = 0.816$). As $p > 0.05$, the null hypothesis of equal distributions across groups was not rejected. This result indicates that there were no statistically significant differences in comprehension scores among the three groups.

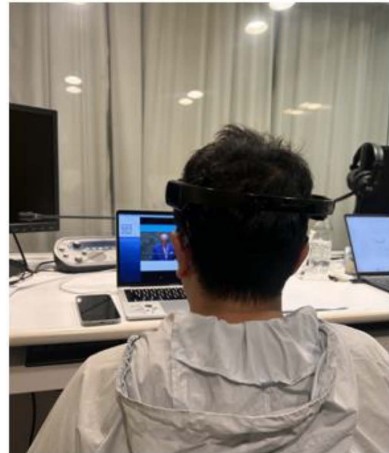 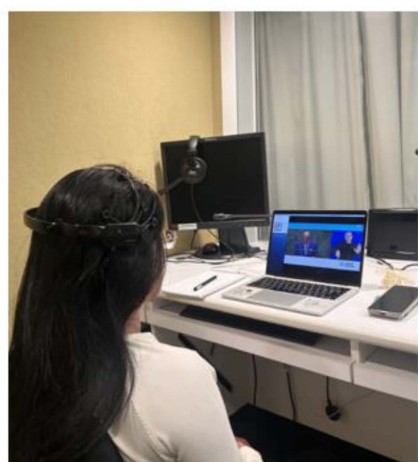

**Fig 2. Participant wearing the EEG device while watching the video.**

Descriptive statistics (Table 2) show a gradual increase in mean scores: Group A (automatic subtitles) scored 70.00, Group B (SI) scored 74.00, and Group C (dual-modality) scored 75.33. Although the differences were statistically non-significant, this incremental pattern suggests potential benefits of dual-modality delivery for comprehension.

Group A exhibited the highest variability ($SD = 23.30$), with scores ranging from 20 to 100, compared to narrower ranges in Group B (40–100) and Group C (30–100). Notably, all groups included participants achieving perfect scores (100), indicating that each modality could support optimal comprehension under favorable conditions. Group C, however, combined the highest mean score with a relatively homogeneous distribution ($SD = 18.85$), suggesting enhanced stability in information processing.

## Cognitive effort

Participants' neurological responses to three distinct content delivery modes were investigated through analysis of brain activity patterns across participant groups. The topographical distribution of the raw EEG data was visualized using the EEGLab toolbox [61] in Matlab R2022a, as illustrated in Figs 3 and 4. The analysis revealed distinct patterns of neural activity across the three groups, with particular emphasis on power bands, which are strongly associated with human cognitive performance.

Fig 3 presents the neural oscillations associated with theta waves, which have been recognized as reliable markers of cognitive effort [49], for the three groups. An increase in theta wave power is typically linked to tasks that demand greater cognitive effort, impose strain on working memory [50], or result in mental fatigue [53]. Moreover, theta waves serve as indicators of mental resource allocation during the processing of multimodal (audio-visual) content [51,52].

**Table 2. Comprehension scores of the three groups.**

| Group | Max | Min | Mode | Mean | SD |
|---|---|---|---|---|---|
| Group A | 100 | 20 | 90 | 70.00 | 23.30 |
| Group B | 100 | 40 | 80 | 74.00 | 15.95 |
| Group C | 100 | 30 | 90 | 75.33 | 18.85 |

Notes: Group A: Automatic subtitles; Group B: SI; Group C: Automatic subtitles + SI.

As shown in Fig 3, Group B demonstrated the highest theta band activation in both the frontal region (AF3) and the occipital region (O7), followed by Group A, with lower theta band activation in the frontal region (AF3), and then Group C, with limited theta band activation.

## Viewing experience

Regarding the viewing experience, Fig 4 illustrates the hierarchical pattern of beta waves associated with active cognitive processing, problem-solving, and alertness, providing insights into the participants' engagement levels and potential stress responses [60,62]. Elevated beta waves activity is generally correlated with excessive levels may indicate anxiety or stress [60,62]. Results show that Group A exhibited the highest beta band activation, indicating the highest stress level, followed by Group B, with Group C showing the lowest level of stress.

In addition to the raw EEG data, which were subsequently drawn as a topographic brain activity heatmap, the Emotiv Pro Suite software package was used to obtain stress values for the participants from the EEG devices while watching the 220-second conference video. We performed the non-parametric Kruskal-Wallis $H$ test for the three groups, and the

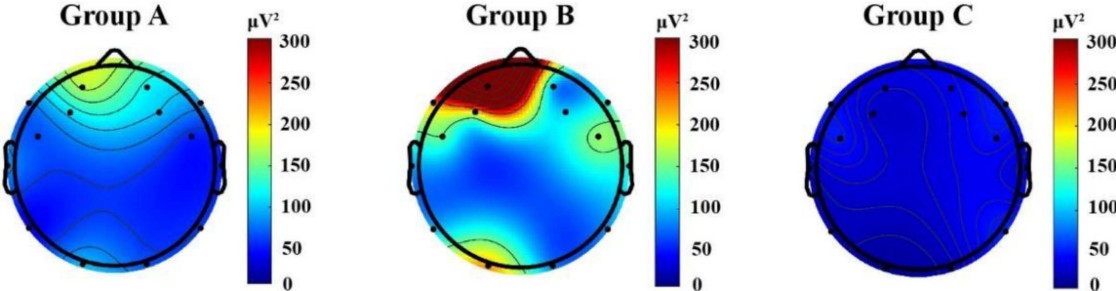

**Fig 3. Topographic brain activity heatmap of EEG data in the theta band.**

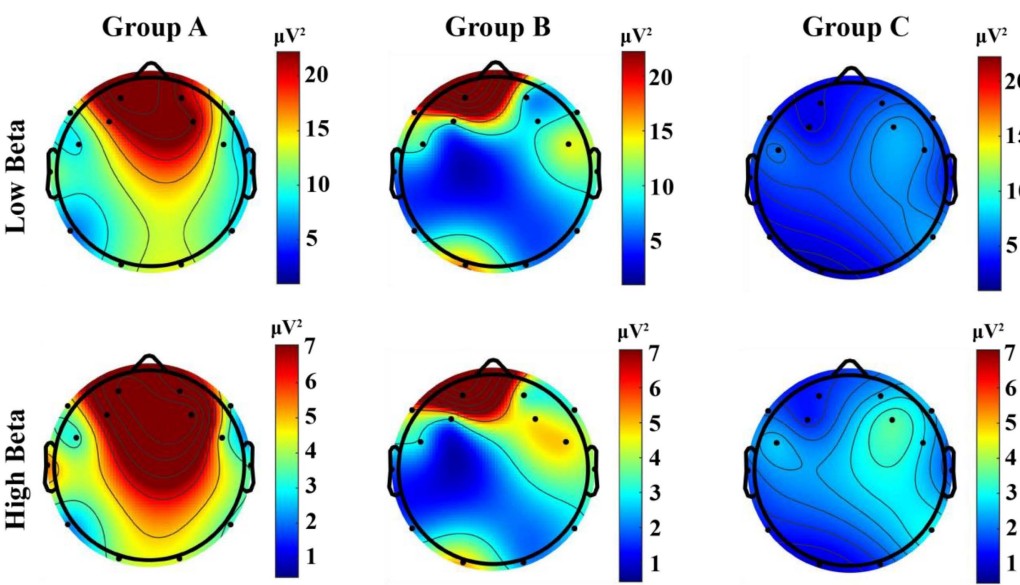

**Fig 4. Topographic brain activity heatmap of EEG data in the beta bands.**

results are shown in Fig 5. There were significant differences in the pressure values of the three groups $p < 0.01$, demonstrating the accuracy and rationality of the discrepancies in the beta band.

## Discussion

### How do automatic subtitles compare to SI in facilitating content comprehension for viewers?

The experimental results revealed that the mode of delivery of translated information across the three groups did not significantly impact viewers' comprehension scores. This aligns with findings by Szarkowska et al. [19], who observed that the absence of audio input had only a modest effect on comprehension when subtitles were available. In addition, Liao et al. [63] found that when participants watched videos with subtitles but without accompanying audio, comprehension scores remained similar to those of participants who had access to both modalities. Other studies have further supported this observation, finding that even the non-syntactic segmentation of subtitles (i.e., splitting sentences at unnatural points) does not negatively impact viewers' comprehension [64,65].

While our statistical analysis revealed only minor differences in comprehension scores across experimental conditions, the group exposed to a combination of automatic subtitles and SI achieved the highest average comprehension scores. This finding highlights the potential of multimodal translation delivery to enhance participants' understanding [19]. By distributing information across auditory and visual channels, the dual-modality approach allows participants to process content more efficiently, as cognitive resources can be shared between modalities, reducing the strain on any single processing channel.

This distribution aligns with the cognitive theory of multimedia learning [66,67], which suggests that synchronizing corresponding visual and auditory material reduces cognitive load. Subtitles can clarify ambiguous or unfamiliar auditory input, while auditory content provides additional context or emotional nuance that subtitles alone may not fully convey [19]. The complementary nature of these modalities enhances understanding while minimizing the risk of cognitive overload, particularly in complex or fast-paced content. Consequently, multimodal delivery systems, such as the combination of automatic subtitles and SI, represent an effective and accessible strategy to support participants' comprehension, especially in multilingual or cognitively demanding environments.

### What differences in cognitive demands do automatic subtitles and SI impose on viewers?

The experimental findings yielded important insights into cognitive effort during conference content processing across the three experimental groups. Analysis of EEG topographic brain activity indicated significant variations in cognitive effort between the SI (listening) and automatic subtitles (reading) conditions, offering valuable theoretical and practical implications.

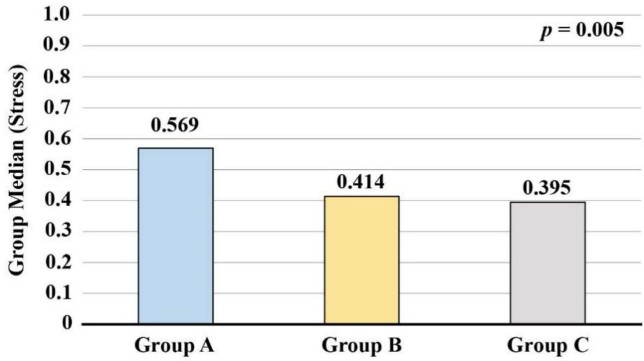

**Fig 5. Non-parametric Kruskal-Wallis *H* test results (stress) for the three groups.**

The theta waves measurements demonstrated a hierarchical pattern of cognitive demand: Group B (SI only) exhibited the highest theta power, followed by Group A (automatic subtitles only), while Group C (combined SI and automatic subtitles) showed the lowest level. This finding is particularly relevant to conference interpreting, as conference attendees relying on translation services experience varying levels of cognitive load depending on the modality through which they receive translated information. The heightened cognitive effort in the SI-only condition suggests additional mental resources were needed for participants to comprehend by processing only auditory information while those listened to SI likely due to the mental effort required to process and comprehend auditory input in real time. In contrast, attendees using automatic subtitles only showed lower cognitive demand, as reading visual text may involve a less resource-intensive processing pathway. Most notably, attendees who had access to both SI and auto-subtitles, exhibited the lowest theta power, indicating reduced cognitive strain when provided with multimodal inputs. The minimal cognitive effort observed for Group C finds robust theoretical support from the bilingual dual-coding theory [68] that suggests memory performance improves when information is processed through multiple channels, in our case the visual and audio channels. These findings extend beyond simple cognitive load measurement, suggesting that the traditional reliance on audio-only SI may not fully optimize audience comprehension and engagement.

The comparison of Groups A and B reveals an interesting deviation from conventional assumptions that watching subtitle videos without corresponding audio would increase cognitive load as viewers use metacognitive strategies to adjust their visual processing to maintain comprehension [19,69]. Audio input is transient and sequential, while visual input is often static and simultaneous, allowing viewers to process information at their own pace and revisit it as needed. Our finding may reflect the experimental setting, which simulated a conference setting where attendees typically listen to speakers for various purposes. In this context, participants were motivated to understand the experimental material, whether through auto-subtitles, SI, or a combination of both, potentially requiring additional cognitive resources for comprehension and retention. Our findings suggest that the professional context significantly may influence how different modalities affect cognitive processing.

### How do automatic subtitles and SI influence viewers' overall viewing experience?

Beta wave analysis revealed distinct stress patterns across the three groups in relation to the viewing experience, providing insights on how different modes of translation delivery impact the overall viewing experience. Elevated beta activity is closely associated with increased mental workload and emotional tension, making it a reliable physiological marker for cognitive strain in audiovisual tasks [55]. Previous studies have employed beta wave analysis to assess user experiences in various applied contexts, such as game design and operational platform usability [46,47].

In this study, participants in Group A (automatic subtitles only) exhibited the highest beta wave levels, suggesting the greatest cognitive load, potentially diminishing subjective viewing comfort. This could be attributed to the demands of processing complex professional content solely through the visual modality. As Szarkowska et al. [19] observed, in the absence of auditory input in the first language (L1), viewers often adopt compensatory metacognitive strategies, such as reading subtitles more thoroughly, which can increase cognitive effort and stress.

Participants in Group B (SI only) showed lower beta wave levels compared to participants in Group A, suggesting a relatively smoother viewing experience. The real-time auditory input in SI likely facilitated reduced cognitive load by conveying prosodic features—such as tone, emotional emphasis, and pragmatic cues—that are inherently absent in text-based automatic subtitles. However, interpreting-based delivery is not without its own constraints. As prior studies have shown [70,71], the transient nature of spoken input places substantial demands on working memory, especially when the content is fast-paced or linguistically dense. Viewers must maintain high levels of attention, with no opportunity to pause, replay, or visually reinforce missed information.

Group C, which combined automatic subtitles and SI, demonstrated the lowest beta wave levels across all groups, suggesting the most cognitively relaxed and effective viewing experience. This dual-modality approach could balance

cognitive load by leveraging the complementary strengths of both modalities. Automatic subtitles serve as a visual rein-forcement, offering viewers extended exposure to verbal content that can support decoding or fill in missed auditory information. Meanwhile, SI enriches the experience by providing expressive and contextual nuances that subtitles may fail to capture. The redundancy in information delivery across modalities allows for greater processing flexibility: when one channel is momentarily ineffective or ambiguous, the other can compensate [66]. This integration not only reduces stress but also enhances comprehension and information retention.

These findings align with the Cognitive Theory of Multimedia Learning [66,67], which posits that when auditory and visual information are temporally aligned, cognitive resources are more efficiently distributed across dual channels. Such integration reduces the likelihood of overload in a single modality and enhances overall processing fluency. While our results do not confirm statistical significance in comprehension performance, the physiological evidence from EEG beta activity strongly suggests that dual-modality delivery promotes a more comfortable and cognitively sustainable viewing experience in demanding multilingual environments.

## Practical implications

The findings of this study offer several practical implications for reducing cognitive load and enhancing the usability of audiovisual content, particularly in contexts demanding SI and real-time accessibility. These insights are particularly relevant for online and hybrid conferences, where addressing the diverse needs of international audiences is critical to ensuring both engagement and comprehension.

First, this study reinforces the critical role of subtitle design in modulating cognitive load. Subtitle systems should not only prioritize linguistic accuracy but also be designed for cognitive accessibility, especially for non-native or linguistically disadvantaged users. In practice, this means ensuring optimal segmentation, timing, and readability, which can alleviate processing burden during complex or rapid content delivery. In multilingual classrooms or international live broadcasts, AI-powered subtitle systems, especially those informed by cognitive feedback metrics such as EEG or user-reported load [72,73], can support more inclusive and cognitively sustainable communication.

Second, the superior performance of the combined modality (SI with automatic subtitles) highlights the value of multimodal delivery in supporting comprehension and reducing stress. In specialized or technical conference settings, where terminology density and speech rate are often high, subtitles act as a visual scaffold, while SI provides pro-sodic and emotional cues that enhance contextual understanding. This dual-modality approach distributes processing demands across auditory and visual channels, thereby reducing cognitive overload and improving the overall viewer experience.

Moreover, although comprehension scores did not differ significantly across conditions, the dual-modality group exhib-ited both lower stress levels and the highest mean performance. These findings suggest that scalable, technology-driven multimodal systems can enhance cognitive resilience and flexibility. In cost-sensitive scenarios or instances where SI is unavailable, auto-subtitling may serve as a provisional alternative, provided that accuracy and synchronization are main-tained. With continued advancements in AI and user-centered design, such systems can be refined to meet the increasing demand for inclusive and efficient multilingual communication in professional and globalized settings.

## Conclusions

This study investigates the combined effects of auto-subtitling and SI on viewers' comprehension, cognitive effort, and overall viewing experience in a conference interpreting setting. The results showed that integrating automatic subtitles with SI reduced cognitive effort and improved the overall viewing experience compared to either modality alone, though comprehension differences were not statistically significant. EEG data revealed that participants in the automatic subtitle-only condition (Group A) exhibited the highest beta activity, followed by Group B (SI-only), while the dual-modality condi-tion (Group C) showed the lowest levels of stress. Interestingly, despite the greater cognitive effort, participants in Group B

rated their viewing experience more positively than those in Group A, suggesting that auditory input may provide additional engagement and relaxation.

Despite these promising results, our study has several limitations. These include a relatively small participant group, a short speech duration, and a focus on a single language pair, which may limit the generalizability of the findings. Future research should address these limitations by incorporating larger and more diverse samples, testing longer and more complex materials, and evaluating dual-modality systems across a broader range of language pairs and cultural contexts. In terms of system development, future research can build on these neurocognitive insights to explore adaptive technologies that integrate real-time EEG monitoring with AI-powered subtitle systems. Drawing inspiration from frameworks such as SUMART [72] and ClinClip's EEG-informed modeling [73], future systems could dynamically adjust subtitle presentation parameters, such as speed, segmentation, and lexical complexity, based on ongoing cognitive feedback. Incorporating large language models (LLMs) with neurophysiological inputs, as shown in recent studies on personalized language learning [74,75], may further support real-time user-centered adaptation. Finally, future work should also consider extending accessibility beyond audio-visual modes by incorporating sign language interpreting.

## Supporting information

**S1 File. Questionnaire in EN.**
(DOCX)

**S2 File. EEG based data.**
(XLSX)

**S3 File. Comprehension data.**
(XLSX)

## Author contributions

**Conceptualization:** Yanlin Li.

**Data curation:** Jiawen Diao.

**Formal analysis:** Yanlin Li.

**Investigation:** Jiawen Diao.

**Methodology:** Yanlin Li, Jiawen Diao.

**Project administration:** Jiawen Diao.

**Resources:** Jiawen Diao.

**Supervision:** Andrew K. F. Cheung.

**Validation:** Yanlin Li.

**Visualization:** Yanlin Li.

**Writing – original draft:** Yanlin Li.

**Writing – review & editing:** Andrew K. F. Cheung.

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
