## [Decision Letter · Decision Letter 0]

7 Apr 2025

PONE-D-25-04673Simultaneous interpreting with auto-subtitling: investigating viewers’ cognitive effort, stress, and comprehensionPLOS ONE

Dear Dr. Cheung,

Thank you for submitting your manuscript to PLOS ONE. After careful consideration, we feel that it has merit but does not fully meet PLOS ONE’s publication criteria as it currently stands. Therefore, we invite you to submit a revised version of the manuscript that addresses the points raised during the review process.Both reviewers have provided detailed and actionable feedback. Please read their recommendations carefully and make sure you address their points. Based on their comments, it would also be useful to thoroughly edit and proof-read the manuscript.

We look forward to receiving your revised manuscript.

Kind regards,

David Orrego-Carmona, Ph.D.

Academic Editor

PLOS ONE

Journal Requirements:

We would like to thank the Research and Innovation Office of the Hong Kong Polytechnic University for supporting the project (Project Code: RKQY). This manuscript was also partly supported by funding from Project number P0046386 of the Department of Chinese and Bilingual Studies of the Hong Kong Polytechnic University.

5. In the online submission form, you indicated that the data underlying the results presented in the study are available from the authors upon request.

6. Please remove all personal information, ensure that the data shared are in accordance with participant consent, and re-upload a fully anonymized data set.

Reviewers' comments:

Reviewer's Responses to Questions

**Comments to the Author**

1. Is the manuscript technically sound, and do the data support the conclusions?

Reviewer #1: Yes

Reviewer #2: Partly

2. Has the statistical analysis been performed appropriately and rigorously? 

Reviewer #1: Yes

Reviewer #2: No

3. Have the authors made all data underlying the findings in their manuscript fully available?

Reviewer #1: No

Reviewer #2: No

4. Is the manuscript presented in an intelligible fashion and written in standard English?

Reviewer #1: Yes

Reviewer #2: No

5. Review Comments to the Author

Reviewer #1: I appreciate the opportunity to review this manuscript. This study offers a timely and methodologically innovative exploration of how automatic subtitling combined with simultaneous interpreting affects cognitive effort, stress, and comprehension in conference settings. By integrating EEG to objectively measure cognitive load alongside comprehension tests, the authors provide robust empirical insights into multimodal information processing. The key finding is that dual-modality delivery reduces cognitive demands without compromising comprehension. This has direct implications for designing accessible multilingual communication systems, particularly in virtual or hybrid conferences. The use of neuroscientific tools to validate human-AI collaboration in interpreting contexts represents a compelling advancement for audiovisual interpreting studies and cognitive science, bridging technology with human-centered outcomes.

While the manuscript is well-written and contributes to the field, I recommend a few minor adjustments to enhance its overall impact:

- The interchangeable use of “auto-subtitles”, “automatic subtitles” and “AI-generated subtitles” creates ambiguity. Choose one and use it consistently throughout the paper to avoid confusion.

- Integrate 2023–2024 studies on AI-driven subtitling (e.g., large language models in real-time transcription) and neurocognitive mechanisms of multimodal processing (e.g., theta-beta coupling in cognitive load).

- It would be valuable for the authors to suggest interesting future research directions based on the findings of this study.

- In the practical implications, discussing how the insights from this study could influence real-world applications, particularly in relation to the viewer’s cognitive well-being, would further enhance the impact of this work.

- It would be valuable for the authors to suggest interesting future research directions based on the findings of this study.

- The resolution of the figures in this manuscript should be improved.

This study makes a compelling case for integrating automatic subtitling with interpreting in multilingual conferences. Its innovative methodology, theoretical contributions, and practical relevance align strongly with the journal’s interdisciplinary focus on technology-mediated communication. While minor revisions are needed to enhance clarity and depth, the core findings are significant and merit publication.

Reviewer #2: Your article addresses pertinent issues and presents valuable ideas. The content is engaging and has the potential to contribute to the field. However, I have some suggestions that could enhance its clarity and depth.

As noted in the commented file, several core concepts are not always clearly contextualised. It would be beneficial to provide clearer definitions/descriptions and background information for these key terms.

While your literature review is a good start, I recommend incorporating more supporting sources throughout the article. Integrating additional studies and references beyond the initial literature review can strengthen your claims and provide a broader context for your findings. The case for the research gap is not convincing enough.

I suggest revisiting your research questions (RQs). This could also clarify the scope of your research.

The final sections of the article could be more concise. Streamlining these parts will help in emphasisiing the key takeaways.

In several instances, the manuscript could benefit from additional details, particularly in the methodology and results sections. Supplying more information regarding your methods, analyses, and the context of your results will enhance the understanding and robustness of your work.

Lastly, using clearer language throughout the article will improve readability. The main thing here might just be avoiding the perception of repetition.

6. PLOS authors have the option to publish the peer review history of their article (what does this mean? ). If published, this will include your full peer review and any attached files.

**Do you want your identity to be public for this peer review?** For information about this choice, including consent withdrawal, please see our Privacy Policy .

Reviewer #1: No

Reviewer #2: No

---

## [Author Response · Author response to Decision Letter 1]

21 May 2025

Please refer to the uploaded file named "response to reviewers".

---

## [Decision Letter · Decision Letter 1]

5 Aug 2025

Simultaneous interpreting with auto-subtitling: investigating viewers’ cognitive effort, stress, and comprehension

PONE-D-25-04673R1

Dear Dr. Cheung,

We’re pleased to inform you that your manuscript has been judged scientifically suitable for publication and will be formally accepted for publication once it meets all outstanding technical requirements.

Kind regards,

Robyn Berghoff, PhD

Academic Editor

PLOS ONE

Additional Editor Comments (optional):

Please note you may disregard Reviewer 3's additional comments re: issues they would like to see addressed before publication. 

Reviewers' comments:

Reviewer's Responses to Questions

**Comments to the Author**

1. If the authors have adequately addressed your comments raised in a previous round of review and you feel that this manuscript is now acceptable for publication, you may indicate that here to bypass the “Comments to the Author” section, enter your conflict of interest statement in the “Confidential to Editor” section, and submit your "Accept" recommendation.

Reviewer #1: All comments have been addressed

Reviewer #3: All comments have been addressed

2. Is the manuscript technically sound, and do the data support the conclusions?

Reviewer #1: Yes

Reviewer #3: Yes

3. Has the statistical analysis been performed appropriately and rigorously? 

Reviewer #1: Yes

Reviewer #3: Yes

4. Have the authors made all data underlying the findings in their manuscript fully available?

Reviewer #1: Yes

Reviewer #3: No

5. Is the manuscript presented in an intelligible fashion and written in standard English?

Reviewer #1: Yes

Reviewer #3: Yes

6. Review Comments to the Author

Reviewer #1: I thank the authors for their revisions. I have no further comments and would be happy to see the paper published.

Reviewer #3: Thank you very much for uploading your manuscript. I have thoroughly studied your paper and other reviewer's opinions and your answers. It deems necessary to appreciate you to do such an invaluable research. There are some issues that I suppose they should be addressed before publishing.

First of all, on page 9 of the paper, the color of the waves and cognitive cases are similar to some degree. Please change them. I have seen in the following parts that you have used more colorful pictures; therefore, this is of no hardship for you.

Secondly, you have chosen Arabic as the preferred language. Why have you not selected other languages and do you and your team have proficiency in this language? Since even the interpreted versions may have some problems. How would have you understood the discrepancies with the correct version? Have you cooperated with any Arabic to Chinese interpreter/translator in this regard?

Thirdly, in the experimental section, you have mentioned that you have prepared subjective questionnaire; however, you have only mentioned one of the questions that you have change the wording. I couldn't find the questionnaire at the end of your paper! Please include it with all the responses.

Last but not least s the case of adding ASR in your keywords since this phrase has been frequently used in your paper.

7. PLOS authors have the option to publish the peer review history of their article (what does this mean? ). If published, this will include your full peer review and any attached files.

**Do you want your identity to be public for this peer review?** For information about this choice, including consent withdrawal, please see our Privacy Policy .

Reviewer #1: No

Reviewer #3: **Yes: ** Neda Patdad

---

## [Editor Report · Acceptance letter]

PONE-D-25-04673R1

PLOS ONE

Dear Dr. Cheung,

I'm pleased to inform you that your manuscript has been deemed suitable for publication in PLOS ONE. Congratulations! Your manuscript is now being handed over to our production team.

Kind regards,

on behalf of

Dr. Robyn Berghoff

Academic Editor

PLOS ONE